# Unsupervised Canine Emotion Recognition Using Momentum Contrast

**DOI:** 10.3390/s24227324

**Published:** 2024-11-16

**Authors:** Aarya Bhave, Alina Hafner, Anushka Bhave, Peter A. Gloor

**Affiliations:** 1MIT System Design and Management, Massachusetts Institute of Technology, 77 Massachusetts Avenue, Cambridge, MA 02142, USA; bhaveaarya09@gmail.com (A.B.); bhaveanushka19@gmail.com (A.B.); 2TUM School of Computation, Information and Technology, Technical University of Munich, Arcisstraße 21, 80333 Munich, Germany; alina.hafner@tum.de

**Keywords:** contrastive learning, momentum contrast, Panksepp seven emotions, canine emotions, unsupervised learning

## Abstract

We describe a system for identifying dog emotions based on dogs’ facial expressions and body posture. Towards that goal, we built a dataset with 2184 images of ten popular dog breeds, grouped into seven similarly sized primal mammalian emotion categories defined by neuroscientist and psychobiologist Jaak Panksepp as ‘Exploring’, ‘Sadness’, ‘Playing’, ‘Rage’, ‘Fear’, ‘Affectionate’ and ‘Lust’. We modified the contrastive learning framework MoCo (Momentum Contrast for Unsupervised Visual Representation Learning) to train it on our original dataset and achieved an accuracy of 43.2% and a baseline of 14%. We also trained this model on a second publicly available dataset that resulted in an accuracy of 48.46% but had a baseline of 25%. We compared our unsupervised approach with a supervised model based on a ResNet50 architecture. This model, when tested on our dataset with the seven Panksepp labels, resulted in an accuracy of 74.32%

## 1. Introduction

Dogs and humans have cohabited as species for over 20,000 years [1,2]. This makes dogs humans’ longest and most loyal evolutionary companion, well before horses and bovines were domesticated. However, even in the modern era, despite thousands of years of coexistence, it is still hard for humans to accurately anticipate a dog’s emotions. While dog therapists and behavioral researchers have demonstrated high accuracy at correctly reading the mood state of a dog, novice owners struggle to correctly understand the behavior of their dog. By creating AI and deep learning applications that interpret facial expressions and body posture from images and videos of dogs, expert knowledge can be made generally accessible to permit everybody to interact with their canine companion and correctly read the mood state of their dog.

Jaak Panksepp, in his research about affective consciousness [3], specifically on core emotional feelings in animals and humans, put forth an argument suggesting that the essence of emotional feelings lies within the evolved emotional action systems in all mammalian brains. He presented a view that makes the tough concepts related to the workings of the brain–mind more accessible, and even proposed that affective feelings might express the neuro-dynamics of the brain systems that produce instinctual emotional behaviors [3]. On the other hand, primary-process affective consciousness is regarded as instinctive and universal for all mammals, so it is easier to investigate in animals. While some of the secondary processes—for example, consciousness of feelings in behavioral decisions—may be assessed using exceptionally prudent learning procedures, special emphasis must be placed on identifying the intrinsic emotional action tendencies of an organism.

He argued that the core emotional feelings are represented in the neuro-dynamic landscapes of various emotional action systems, like SEEKING, FEAR, RAGE, LUST, CARE, PANIC and PLAY. By studying these systems, researchers can uncover the neural basis of emotional consciousness in both humans and animals. This approach encourages the ethological analysis of emotional action tendencies and the accompanying brain changes for the effective monitoring of emotional states. Additionally, it suggests that meaningful progress in this field requires more open discussions among animal brain researchers.

The study of animal emotions, known as affective biology, is becoming increasingly popular across various research fields, like evolutionary zoology, affective neuroscience and comparative psychology [4,5,6,7]. From 2014 to 2022, scientists used advanced technologies to understand and improve the emotional well-being of animals by tracking their movements and recognizing their gestures. For instance, Broomé et al. [8] surveyed through research that uses computer vision to assess animal emotions and recognize signs of pain by closely analyzing facial expressions and body language. Identifying animal emotions is hard because they might hide their internal feelings. Traditionally, researchers would watch or record videos of animals to study their behavior, but now, the automatic analysis of facial expressions and body poses allows for a more detailed understanding of their emotional states. Studies on animal emotions have included estimating poses and using deep learning to identify and track animals. Understanding animal emotions by analyzing their facial expressions and body language is more complex than simply tracking their movements.

Lately, scientists have used computer vision and deep learning to recognize emotions in dogs. Hussain et al. [9], for example, used different sensors to capture dogs’ movements and combined different streams of data to check for the well-being of dogs. Similarly, Franzoni et al. [10] conducted experiments to evoke emotions in dogs and focused on detecting those emotions through their facial expressions. Ferres and colleagues took a different approach, recognizing emotions from the body poses of dogs by pinpointing the key areas on their bodies and faces [5]. Most of these studies could not attain much accuracy without proper visibility of dogs’ faces and limbs [4]. Research on dog emotion recognition using computer vision and deep learning has mainly centered around clear, high-resolution images of individual dogs. Surveillance cameras have commonly been used, and the emotional states of the animals have mainly been inferred from their physical behavior. In contrast, past research on human emotion recognition has used text, audio, or video data along with various models to achieve high accuracy, often relying on facial expressions or body language analysis as the input for supervised or unsupervised learning. However, we are currently not aware of any studies using unsupervised learning for dog emotion recognition.

Semi-supervised learning has emerged as a promising approach for leveraging large amounts of unlabeled data to enhance the performance of learning-based networks. By incorporating both labeled and unlabeled data, semi-supervised learning can improve the generalization capability of models. However, if the semi-supervised learning policy is inefficient, then the quality and reliability associated with the learned features does not hold. This basically implies that designing an efficient semi-supervised learning strategy and, hence, ensuring the reliability of the learned features for every specific application, is still an open challenge.

Most facial emotion recognition methods heavily rely on supervision, making it challenging to analyze emotions without considering individuals. Conversely, self-supervised learning offers a way to learn representations without supervision. Kim et al. introduced a new adversarial learning approach [11]. Specifically, it helps a facial emotion recognition network to better understand complex human emotional elements by learning weak emotion samples from strong ones in an adversarial manner. Their method is able to recognize human emotions independently of individuals, leading to a more accurate facial expression understanding by proposing a contrastive loss function to improve the efficiency of adversarial learning. In this paper, we develop a system to recognize emotions in dogs based on their facial expressions and body posture using the contrastive learning framework MoCo [12] (Momentum Contrast for Unsupervised Visual Representation Learning) to identify the seven Panskepp emotions.

The chief contributions of this paper are as follows:We construct a novel, high-quality, diverse and unskewed dataset of 2184 images consisting of the ten most popular dog breeds worldwide, with varying shapes and sizes. These dog breeds are ‘Siberian Husky’, ‘Rottweiler’, ‘Golden Retriever’, ‘Labrador Retriever’, ‘Pug’, ‘Poodle’, ‘Beagle’, ‘German Shepherd’, ‘Pembroke Welsh Corgi’ and ‘French Bulldog’. We construct equally sized groups of these images for the seven Panskepp emotion labels “Exploring”, “Sadness”, “Playing”, “Rage”, “Fear”, “Affectionate” and “Lust”.We leverage the contrastive learning frameworks SimCLR (a Simple Framework for Contrastive Learning of Visual Representations) and MoCo (Momentum Contrast for Unsupervised Visual Representation Learning) on our dataset to predict the seven Panksepp emotions using unsupervised learning. We significantly modify the MoCo framework to obtain the best possible results on our hardware. We also test the unsupervised learning models on a publicly available dog emotion dataset to compare their relative performance on baseline accuracies.We build a supervised learning model based on a ResNet50 architecture and run it on our dataset as well as the publicly available dataset to obtain benchmark results.

Our research makes significant contributions to the field of Animal–Computer Interaction (ACI) by introducing a novel method for recognizing dog emotions, fostering improved human–animal communication, adhering to the highest ethical standards and utilizing a unique contrastive learning approach. By better recognizing dog emotions, we help bridge the human–animal communication barrier. Additionally, we advance the field of applied machine learning by demonstrating the effectiveness of semi-supervised and self-supervised learning techniques for animal emotion recognition.

## 2. Method

In this experimental study and the subsequent reconfiguration of framework architectures, we have pursued an approach which prioritizes novelty. Consequently, we have prepared an original dataset to carry out this academic research. Following this, we have experimented with contemporary contrastive learning frameworks. A detailed taxonomy of contrastive learning methods was conducted by Le-Khac et al. [13], differentiating among supervised, semi-supervised and unsupervised contrastive learning techniques. After experimentation with several contrastive learning frameworks, we have implemented a modified version of the MoCo framework [12] developed by He et al.

### 2.1. Creating the Dataset

To implement unsupervised visual representation learning [14] on image data, we embarked on the task of collecting, analyzing and evaluating images of dog emotional behaviors [15]. Our key focus was capturing the respective physiological displays for different emotional temperaments among several different dog breeds. The dataset we constructed includes images of the top ten most popular dog breeds worldwide [16], categorically distributed according to their breed and seven different emotional behaviors. These emotions included ‘Exploring’, ‘Sadness’, ‘Playing and Happy’, ‘Rage’, ‘Fear’, ‘Caring and Affectionate’ and ‘Lust’. The ten dog breeds included in this dataset are ‘Siberian Husky’, ‘Rottweiler’, ‘Golden Retriever’, ‘Labrador Retriever’, ‘Pug’, ‘Poodle’, ‘Beagle’, ‘German Shepherd’, ‘Pembroke Welsh Corgi’ and ‘French Bulldog’. In addition to this, we also added a miscellaneous section with images of other dog breeds for reasons discussed further in this section. Interventionary studies involving animals or humans, and other studies that require ethical approval, must list the authority that provided approval and the corresponding ethical approval code.

To ensure that the dataset obtained from Google Images and Wikipedia was representative and unbiased, we chose around 200 images of each breed and divided them into ten popular breeds and one miscellaneous category. To handle emotional bias, we categorized these images into seven emotions: Exploring, Sadness, Playing, Rage, Fear, Affectionate and Lust. Maintaining strict criteria for labeling each image, paying attention to facial expressions, body posture and other non-verbal cues, were factors that contributed to the consistency in their labeling. The images were taken in various environments, such as indoor, outdoor and lighting conditions that may have been more robust, and resulted in the dataset reflecting broadly diversified real-world scenarios and different breeds of canines.

#### 2.1.1. Methodology for Labeling Images

We adhered to the comparative cognition research principles posited by Kujala [17] for analyzing and evaluating our images. The visual identification of dog emotions relies heavily on a detailed observation of facial expressions, body postures and other non-verbal cues. Facial expressions, such as ear position, eye shape and mouth movements, are critical indicators of a dog’s emotional state. For instance, a dog with relaxed ears, soft eyes and a slightly open mouth is typically expressing contentment or relaxation, whereas pinned-back ears, a furrowed brow and a closed mouth might indicate fear or anxiety.

The tail position and movement also provide significant emotional cues; a wagging tail generally suggests excitement or happiness, but the speed and height of the wag can modify the interpretation, for example, a slow wag with a low tail often signals uncertainty or submissiveness. In addition to facial expressions and tail movements, body posture plays a crucial role in the visual identification of emotions in dogs.

A confident and happy dog will usually have a loose, relaxed posture, often with a playful stance or a wagging tail. In contrast, a dog experiencing fear or aggression might exhibit a tense, rigid body, with its weight shifted backward or forward, depending on whether the dog is preparing to flee or confront. Submissive dogs may crouch low to the ground, tuck their tails between their legs and avoid direct eye contact. By carefully observing these visual cues in various contexts and interactions, owners and researchers can gain valuable insights into the emotional states of dogs, allowing for more effective communication and stronger human–animal bonds. Using the above key factors as described in [17], we were able to successfully classify the images into seven emotions as discussed in the section that follows.

#### 2.1.2. Dataset Description

We primarily used Google Images and Wikipedia Images to collect a total of 2184 images of different and distinct individual dogs across the internet, which we collected and manually labeled using the methodology highlighted in the previous section of this paper. It is important to note that the 2184 images were collected from the internet independently of each other, and this was before applying any augmentation to these images for training. It is also important to signify that the image augmentation techniques discussed in the latter sections of this paper did not increase or decrease the size of the original dataset of 2184 images in any way, shape or form. For the sake of preserving copyright, the dataset that we collected cannot be made public for open-source usage, since it contains several images that have copyrights associated with them. Thus, we cannot publicly release them to the scientific community. The search engines were used from 18 January 2024 to 4 April, across a span of three months. The emotional behaviors we considered are adapted from Panksepp’s seven core emotional systems [3].

Jaak Panksepp is one of the most influential neuroscientists in recent times. Panksepp argued that each of these seven emotions has a dedicated system in the subcortical regions in the brain of all mammals. Dogs were domesticated from now-extinct wolves between 11,000 and 16,000 years ago [2]. This provided evolutionary reasons for dogs and humans alike to share and adapt to each other’s emotional behaviors, due to our closely bound symbiotic relationship. The images from this dataset show the seven Panksepp emotions. The emotions ‘Sadness’, ‘Exploring’ and ‘Lust’ were found to be universally exhibited by all dog breeds in equal frequencies, while the emotions ‘Caring and Affectionate’, Playing and Happy’, ‘Rage’ and ‘Fear’ were found to be exhibited in varying frequencies among the different dog breeds. Working dog breeds, like German Shepherds and Rottweilers, tend not to display fearful behaviors as often as breeds like Pugs and Pembroke Welsh Corgis. Similarly, breeds like the Golden Retriever and Poodle rarely exhibit aggressive behavioral tendencies [15].

These factors influenced the image count for each emotion and breed. To tackle this issue, we introduced a ‘Miscellaneous’ category that contains images of other dog breeds that are not considered in the ten breeds listed above. The dataset that we assembled consists of 2184 images, with 300 images per emotion. Approximately 200 images are included for every considered dog breed, also including the miscellaneous category. We also made certain that each image contained the entire body of the subject, along with the environment in which the subject was present. We ascertained that the behavioral displays were largely dependent on the overall stance and body posture of the dogs [12]. The temperament of the dogs varied with respect to their surrounding terrain, the amount of natural lighting and weather conditions [15]. Figure 1 and Figure 2 explore this in detail.

### 2.2. Supervised Learning Benchmarks

ResNet50 is a landmark in deep learning, not only because it enables the training of extremely deep neural networks, but also because its information can bypass multiple layers through residual connections. This architectural advancement has significantly improved the outcomes for many tasks, for example, image classification and object detection.

The ResNet family offers models of various depths for scenarios with limited computational resources: ResNet18, ResNet34 and ResNet50. Applying a smaller ResNet variant allows a user to obtain comparable results but with a significantly decreased demand on computational resources. ResNet18 is a reasonable choice since it can handle the processing of smaller datasets with limited hardware with balanced performance and efficiency. Which variant of ResNet to choose depends on the task, the size of the dataset and, possibly, the availability of computation resources.

We trained a supervised CNN built on a ResNet50 architecture. Overall, ResNet50 performs quite well on image classification tasks, although in direct comparison to other image classification models, such as Alex Net and GoogleNet, it gets confused by certain objects, such a dogs and deer, classifying them as horses [18]. The model utilizes the PyTorch-Lightning framework based out of PyTorch 2.2.1, to streamline the training and validation processes. The image data were preprocessed and augmented through a series of transformations to enhance the model’s generalizability and to mimic a variety of real-world conditions. These transformations included random rotations of ±10 degrees, horizontal flips with a 50% probability, resizing images to 224 × 224 pixels, center cropping and normalization using the ImageNet dataset’s mean and standard deviation values ([0.485, 0.456 and 0.406] and [0.229, 0.224 and 0.225]).

The dataset comprises 2184 images categorized into 7 classes representing the Panksepp’s emotions [3] (i.e., ‘Exploring’, ‘Sadness’, ‘Playing’, ‘Rage’, ‘Fear’, ‘Affectionate’ and ‘Lust’), with the directory structure facilitating the use of PyTorch’s ImageFolder for automatic labeling based on the folder names. A custom DataModule class, an abstraction provided by PyTorch Lightning, managed the data loading and splitting. The dataset was divided into an 80-20 ratio for training and testing. For each subset, ‘DataLoaders’, which is a PyTorch sampling library, shuffled the training data and batched them into sets of size 32, optimizing the loading process and preparing the dataset for efficient training. The core of our model is a CNN based on a ResNet50 architecture, and pretrained on the ImageNet dataset.

Transfer learning was employed, leveraging the pretrained convolutive bases to extract features, while the fully connected output layer was adapted to our specific task of classifying emotional states. This output layer was redefined to match the number of classes in our dataset, replacing the original ImageNet classifier. The CNN utilized an Adam optimizer with a learning rate of 0.001. The training involved the computation of cross-entropy loss, a common choice for multi-class classification problems. The performance metrics, specifically loss and accuracy, were logged and monitored during the training to assess the model’s convergence and effectiveness on both the training and validation datasets. Using PyTorch Lightning’s Trainer, the model underwent 20 epochs of training, where it learned to minimize the loss function and improve its accuracy on the processed images. The framework handled the under-the-hood functionalities like GPU acceleration (if available) and model checkpointing, allowing us to focus on the model architecture and performance.

### 2.3. Unsupervised Learning with Contrastive Learning Frameworks

In computer vision research, novel sophisticated methodologies, such as contrastive learning and unsupervised visual representation learning, have initiated a paradigm shift in representation learning and pretraining strategies. Contrastive learning, while originally developed within the framework of unsupervised learning, has demonstrated remarkable potential for few-shot learning scenarios [19]. Contrasted learning precisely focuses on forcing a model to push closer representations of connected images, rather than pushing apart representatives of the least-related images in an embedding space. The process typically involves constructing pairs or batches of images and applying augmentation techniques to create different views of the same image. These views are then passed through a neural network to generate embeddings [20]. In the context of emotional behavior analysis, this means contrasting instances of similar emotional expressions (positives) while distinguishing them from dissimilar expressions (negatives).

#### 2.3.1. Why Contrastive Learning?

Emotional behaviors often exhibit complex and subtle variations influenced by factors such as cultural background, individual differences and contextual cues. Contrastive learning can help disentangle these variations by forcing a model to focus on the discriminative features that distinguish between different emotional states, thus enabling more nuanced and robust representations [21]. The emotions chosen for the dataset were such that they produced diametrically opposite physiological displays among the dogs. Thus, a contrastive learning framework should be able to produce well-separated plots for all the labels within an embedding space. Additionally, this opens the possibility of finding novel emotional behaviors that may be represented as outgrowths of an existing label cluster. The cluster formation and corresponding clustering methods used were motivated by Zhang et al. [22].

In addition, several studies have demonstrated the effectiveness of contrastive learning for learning rich representations from images. For example, Jaiswal, A. et al. [23] conducted a survey of contrastive learning methods and their applications, highlighting the ability of contrastive learning to learn representations that capture semantic information in images with good accuracy metrics. In conclusion, contrastive learning is a powerful technique that can be used for the analysis of emotions using image data. By learning to differentiate between images that represent different emotions, contrastive learning can capture the unique visual features associated with each emotion. This makes it a valuable tool for emotion discovery and analysis in scenarios where labeled data are limited, as it can learn meaningful representations from unlabeled data, as demonstrated by Shen et al. [24].

#### 2.3.2. Experimenting with the SimCLR Framework

The Simple Framework for Contrastive Learning of Visual Representations, developed by Chen T. et al. [25], is a framework which offers effective representation learning for medium-to-large-scale datasets. Its novelty lies in the fact that it is the most straightforward contrastive learning framework that still manages to outperform several other more complex frameworks. In our implementation of the SimCLR framework, we have utilized a pretrained ResNet50 as the encoder. We have used a non-linear projection head that includes a fully connected layer with 2048 input features and 2048 output features, a ReLU activation layer and another fully connected layer with 2048 input features and 128 output features.

In addition to the modifications in the architecture of the multi-layer perceptron, we have also made some changes to the pretrained ResNet50 encoder. We have replaced the parameters for the first convolutional layer, changing the kernel size to (3, 3), the stride to (1, 1) and removed the padding. The minimum recommended batch size in the SimCLR [25] paper is 256, with the optimum batch size being 2048. We have used the Layer-wise Adaptive Rate Scaling (LARS) optimizer put forth by You Y. et al. [26]. We have used a learning rate of 0.2 and a weight decay of 10-6.

We have also implemented a linear warmup of 10 epochs, after which we have followed with a cosine annealing learning rate scheduler for 500 epochs. We have used an input image size of 32 × 32 in order to be able to train with an NVIDIA RTX 3070 Ti GPU with 8 Gigabytes of video memory. We have used the Normalized Temperature-scaled Cross-Entropy Loss (NT-Xent) from [27] to train the encoder. The temperature value we have used for the NT-Xent loss is 0.5. Because of the resource-intensive nature of the SimCLR framework, we have not implemented a downstream classifier and instead have directly generated a t-distributed stochastic neighbor embedding with a perplexity value of 50.

#### 2.3.3. Frameworks That Require Less GPU Memory

Contrastive learning frameworks have an intrinsic need for a large batch size, due to the fact that a lot of negative samples are required to properly differentiate data points from each other within an embedding space. The SimCLR [25] framework has a recommended batch size of 2048, which is not implementable with the memory sizes of most GPUs.

Momentum Contrast for Unsupervised Visual Representation Learning [12], also known as MoCo, is another popularly used contrastive learning framework that utilizes a technique called ‘Cross Batch Memory Accessing’ [28] to get around the issue of requiring a large batch size. The MoCo framework involves the implementation of a queue to store the features generated by past batches, and uses them as negative samples during training. The MoCo frameworks uses two encoders unlike SimCLR, which uses a single encoder interchangeably. The two encoders are called ‘Query Encoder’ denoted by ‘encQ’, and ‘Momentum Encoder’ denoted by ‘encK’. The query encoder is updated using a stochastic gradient descent, while the momentum encoder is updated using an exponential moving average.

Another efficient way of resolving the issue of GPU memory is ‘Model Parallelism’. Model parallelism involves splitting a model across multiple GPUs, with each GPU responsible for computing a subset of the model [29]. This allows for the training of larger models that would not fit into a single GPU’s memory. However, model parallelism introduces communication overhead between the GPUs, which can impact the training performance.

In addition to this, we also considered Minibatch Gradient Checkpointing (MBGC). MBGC is a framework that enables training large models with limited GPU memory by checkpointing activations and recomputing them during a backward pass to reduce memory usage [30,31]. It divides a model into segments and checkpoints intermediate activations, thereby trading off computation for memory. This approach allows for the training of larger models without significantly increasing the memory footprint.

### 2.4. Momentum Contrast for Unsupervised Visual Representation Learning

Among the frameworks that use Momentum Contrast are MoCo [12], MoCov2 [32] and MoCov3 [33]. The original MoCo framework is the computationally lightest framework to train. For this project, we implemented a modified version of the original MoCo framework.

The modifications include the alteration of the data augmentations, the use of an alternate encoder and the slight changing of the encoder architecture. The most notable change we made are the data augmentations.

The MoCo [12] paper suggested the use of different data augmentations. After testing its performance with different augmentations, we found that the following augmentations produce the best results.

#### 2.4.1. Data Augmentations

The following data augmentations were performed on the training data:Random resizing and cropping.Random horizontal flip (probability = 0.5).Random application of color jitter with the jitter values for brightness = 0.4, contrast = 0.4, saturation = 0.4, hue = 0.1 and probability = 0.8.

As demonstrated in MoCo [12], we built our training, test and validation data loaders. The image data augmentations were applied to the same image twice by the training data loader, converted to a PyTorch tensor, normalized and fed into the model for training as two positive samples.

These augmentations had the best effect for increasing the diversity of the dataset and encouraging the model to learn embeddings efficiently from our training data loader. The augmentations were initially borrowed from the MoCo repository [12], but were modified according to their effect on the model’s performance. The size of the image fed into the model was treated as a hyperparameter during training, due to its effect on the GPU’s memory consumption and accuracy. Figure 3 explores the image augmentation.

#### 2.4.2. Modification of ResNet34 and ResNet18 Architectures

The MoCo paper discussed experimentation with R50w4x, R50w2x and R50 as encoders. These encoders are resource-intensive and difficult to train with limited hardware. For this project, we had to experiment with ResNet34 and ResNet18 as the encoders. We implemented the commonly used CIFAR-10 ResNet18/34 recipe, which, in comparison to the ImageNet ResNet recipe, does the following:Replaces the first convolution layer with a kernel size = 3 and stride = 1.Removes the first pooling layer.

In addition to these changes, each batch normalization layer was replaced with split batch normalization layers. Split batch normalization [34] is a technique that allows for the simulation of multiple GPUs behaviors on a single GPU. Each split batch normalization layer was set to carry out 8 splits across a batch. We also used Data Parallelism [35] in order to train these split batches on a single GPU.

The query encoder was updated using stochastic gradient descent, and the NT-Xent loss function was found to better perform on this project than the Info NCE loss function.
$$L_{q}=-\ln\left(\frac{e^{\left(\frac{q\cdot k_{+}}{\tau }\right)}}{\sum_{i=0}^{k}e^{\left(\frac{q\cdot k_{i}}{\tau }\right)}}\right)$$

The momentum encoder was updated using an exponential moving average as used in the MoCo [12] paper. The ResNet18 as well as the ResNet34 encoders were set to produce final feature vectors of 128 in length.

### 2.5. KNN Classifier

The feature vectors produced by the encoder were tested for accuracy using the KNN classifier borrowed from [36]. The value for the ‘K’ nearest neighbors to be considered in the embedded space was set to 200. The classifier used the cosine similarity as the distance metric for the feature vectors. The implementation of this was taken from the GitHub Repositories http://github.com/zhirongw/lemniscate.pytorch (accessed on 21 March 2024) and https://github.com/leftthomas/SimCLR (accessed on 21 March 2024). The learning rate scheduler was a cosine annealing scheduler with the value for temperature set to 0.1, since it performed best with this InstDisc KNN monitor [36].

It is important to note that a fully connected downstream network that uses the pretrained encoder features as the input will produce an outstanding accuracy for practical tasks. The similarity scores are adjusted by a temperature parameter and converted to probabilities. Next, the classifier creates a one-hot encoding for the labels of the nearest neighbors and calculates the weighted sum of these one-hot vectors to obtain the predicted scores for each class.

The KNN monitor in InstDisc [36] serves as a mechanism with which to utilize the learned representations for identifying potential negative pairs, which enhances the quality of the learned representations for downstream tasks. The choice of K in the KNN monitor is a key hyperparameter that can be tuned to balance the diversity and selectivity of the negative pairs.

### 2.6. Final Network Architecture

Figure 4 is a simple representation of the Momentum Contrast framework that we have utilized in this study.

This architecture, designed for canine emotion recognition, utilizes a self-supervised contrastive learning framework with two ResNet encoders, trained on a curated dataset of dog images sourced from Google Images and Wikipedia, covering a diverse range of breeds and emotional states. The model takes two augmented views of each image as the inputs to capture variations in the visual representation—each view represents a modified version of the same image, applying transformations like cropping, color adjustments and rotation. These views are fed into two separate ResNet encoders: a query encoder, which is updated via stochastic gradient descent (SGD), and a momentum encoder, which is updated using an exponential moving average (EMA) of the query encoder’s weights, providing stability across the training steps. The encoders transform each view into high-dimensional feature representations. By measuring the cosine similarity between the two outputs, the model is trained to bring the representations of similar images (different views of the same image) closer together in the feature space while pushing apart the representations of different images. This similarity is enforced through the NT-Xent loss (Normalized Temperature-scaled Cross-Entropy Loss), which helps the model distinguish between the different emotional expressions of dogs by learning the invariant and discriminative features across various breeds, emotions and environmental contexts, ensuring robustness and generalization across real-world conditions.

## 3. Results

Throughout this project, we have made extensive use of PyTorch, which is a deep learning and hardware acceleration library in Python v.3.9. While experimenting locally with our NVIDIA RTX 3070-Ti GPU, we have used Ubuntu-22.04 LTS, which is an open-source Linux-based operating system built by Canonical, London, England. While training on the NVIDIA Tesla P100 GPU, we have used Kaggle’s default shell for operating the Linux commands during training and analysis. Locally, we have used an Intel-12800HX CPU, manufactured by Intel, Santa Clara, CA, United States, and in Kaggle’s remote environment we have used their Intel-Xeon CPU. It is important to note, however, that the CPU makes little difference for the outcome of our experiments.

In this section, we explore the performance of our unsupervised learning models with different values for the hyperparameters and different training environments, and compare them with two supervised learning models that use a simple CNN based on a ResNet50 architecture. For the hyperparameter values, we have experimented with the batch size, the value of K in the KNN monitor classifier [36] and the number of epochs for which the model was trained. We have used a modified SimCLR [25] framework and several different modified MoCo [12] frameworks for the unsupervised representation learning in this project. All the relevant code and results will be available in GitHub [link removed for anonymized review].

### 3.1. Results Produced by the SimCLR Framework

The SimCLR [25] framework was used in the initial exploratory steps for contrastive learning in this project. The maximum resolution that could be used with this framework was 32 × 32, paired with a batch size of 256. The computational intensity of the SimCLR framework prevented us from training it on our dataset with seven emotion labels. As a result, we attempted to train the model using a publicly available dataset, put together by Daniel Shan Balico on Kaggle [37]. Despite being trained on the publicly available dataset with only four labels [37], the model failed to attain an accuracy of 30%, where the base level guessing accuracy would be 25%, since only four labels are available in [37]. The implementation of that can be viewed in our repository. This made the model impractical to proceed with and to build downstream networks.

### 3.2. Results Produced by the MoCo Framework

All the MoCo models were trained on our own proprietary dataset with the seven Panksepp emotion labels. The baseline accuracy for this dataset was 14.28%, and our unsupervised learning models managed to achieve an accuracy of 40%. Table 1 shows the hyperparameter values and the performance metrics for our best-performing models.

#### Testing Accuracy and Training Loss for Our Unsupervised Learning Models

In the following section, we display our test accuracy and training loss graphs. In all of the training runs, we have seen a plateau appear after 700 epochs. We have observed more variance in the accuracy values when training with the more resource-intensive ResNet34 encoder. Figure 5, Figure 6, Figure 7 and Figure 8 explore the performance of our model.

Figure 9 compares the performances shown by the different unsupervised models and the supervised control model on our dataset.

### 3.3. Comparison of Supervised and Unsupervised Results

Post-training, the supervised model described in Section 3.2 was evaluated on two datasets: a separate test dataset to measure its generalization capabilities and our Panksepp seven emotions dataset. The classification report, provides detailed insights into the model’s performance across all seven Panksepp emotional categories. The model achieved high precision for the ‘Rage’ category (100%), indicating that all the predictions for this class were correct, though the recall suggests that it missed a significant portion of the actual ‘Rage’ cases. Conversely, the ‘Fear’ class showed an impressive recall of 94.29%, suggesting the model is highly sensitive at identifying this emotion but less precise, as indicated by a lower precision of 51.56%. This implies a higher number of false positives for ‘Fear’.

The overall accuracy of the model stands at 74.32%, which reflects a solid ability to generalize across unseen data. The macro average F1-score of 74.90% suggests a balanced mean performance across classes, accounting for the imbalance in the class distribution as indicated by the ‘support’ values. However, the disparity in performance among the different emotions suggests that while the model is adept at recognizing certain emotional states (like ‘Playing’ and ‘Rage’), it struggles with others, such as ‘Sadness’ and ‘Exploring’.

Comparing the supervised learning (see Table 2) scores to the results achieved by our approach based on the MoCo framework, we conclude that the contrastive learning results are promising, especially for predicting ‘Lust’, followed by ‘Rage’ and ‘Sadness’. ‘Caring’, however, compared to the supervised learning approach, seems to be harder to predict. Table 3 shows the classification report in accordance with ResNet50 and its performance.

Furthermore, we tested our CNN architecture on a publicly available dataset [37] containing 800 images representing each of the four emotional states ‘Angry’, ‘Happy’, ‘Relaxed’ and ‘Sad’. Again, a classification report, shown in Table 4 and Table 5 provides detailed insights into the model’s performance across all the emotional categories.

The CNN demonstrated a strong performance across all the classifications, as evidenced by the precision, recall and F1-score metrics for each category. Specifically, it achieved the highest precision for the ‘Angry’ emotion at 93.79%, indicating a high degree of accuracy at identifying this specific state. However, its recall for ‘Angry’ was lower at 77.04%, suggesting that some instances were missed. The ‘Happy’ emotion scored impressively on both precision (87.85%) and recall (94%), leading to the highest F1-score of 90.82% among all the categories. The ‘Relaxed’ and ‘Sad’ emotions also showed robust results, with ‘Sad’ exhibiting a notably high recall rate of 91.30%. Overall, the model achieved an accuracy of 85.12% on the dataset of 800 images, with the macro and weighted averages of the precision and F1-score at around 85%. This indicates a consistent and balanced performance across the different emotional expressions.

Again, comparing the supervised learning results (see Table 4) to the results achieved by our approach based on the MoCo framework, we can again conclude that the contrastive learning results are promising, especially for predicting ‘Angry’ and ‘Sad’ dogs, but still significantly less accurate than the supervised learning results.

Comparing the different variants of the MoCo [12] framework, we found that contrastive learning showed a promising performance on the analysis of emotional behaviors in dogs. In comparison to the ResNet50 architecture, which used 23.9 million parameters, the ResNet18 and ResNet34 encoders used only 11.4 million and 21.5 million parameters. The primary requirements of any contrastive learning framework are a large batch size and a heavy encoder. In spite of this, the models that we built showed promising results with their lighter encoders and computationally limited training environments [38].

### 3.4. Generalizability of Our Results

Our results demonstrate the strong potential of contrastive learning frameworks, particularly the MoCo variants, for analyzing emotional behaviors in dogs. The effectiveness of our models, despite using lighter encoders and operating within computational constraints, underscores the adaptability and robustness of these approaches. However, it is essential to recognize the limitations posed by the specific dataset and the potential variability in real-world scenarios. Future research should focus on evaluating these models across diverse datasets, including those representing different breeds and environments, to ensure broader applicability. Additionally, exploring transfer learning techniques could enhance the generalizability, allowing models trained on dog emotion datasets to be adapted to other species or even human emotional recognition tasks. This cross-species application could open new avenues for understanding and interpreting emotional behaviors using machine learning, fostering advancements in both animal behavior research and human–animal interaction studies.

The potential implications of the generalizability of our results are significant. If these models could be effectively applied across diverse datasets, it could lead to a more accurate and nuanced understanding of dog emotions in various contexts, such as different breeds and environments. This could enhance the development of applications in areas such as animal welfare, training and therapy. Moreover, exploring transfer learning techniques could enable the adaptation of these models to other species or human emotional recognition tasks. Such cross-species applications could advance our understanding of emotional behaviors more broadly, facilitating improvements in human–animal interaction studies and potentially contributing to the development of better tools for emotional recognition and response in both animals and humans.

Our current model relies simply on facial expression and postures. It is thus possibly biased toward a restricted ability to be accurate while distinguishing between several emotions, as canine emotions also depend upon environmental factors or social information, such as the presence of toys, animals and even humans. Therefore, future experiments with contextual integration through training on rich datasets with diverse backgrounds that can encompass a variety of interactions will increase accuracy. This could be made possible by integrating contextual cues through multimodal approaches or context-conditioned training, so that the model would better interpret and differentiate subtle emotional differences in dogs.

### 3.5. Misclassification Within Our Results

Among the unsupervised model’s results, specific misclassifications revealed critical insights into the challenges of canine emotion recognition using contrastive learning frameworks. Misclassifications occurred primarily among emotions with similar visual features, suggesting that the subtlety of certain emotional expressions in dogs may lead to an overlap in feature representations.

(i).Exploring vs. Playing:

These emotions were commonly confused, likely due to the shared indicators of alertness and active body posture. Both involve forward-facing stances, bright or widened eyes and upright ears, which might be challenging for the model to distinguish without contextual cues (e.g., environmental background indicating exploration or a toy indicating play).

(ii).Fear vs. Sadness:

Another area of confusion was between fear and sadness, where both emotions show a certain degree of lowered body posture and backward-facing ears. The model may benefit from additional preprocessing that emphasizes the subtle differences in facial tension or tail positioning that separate these two emotional states.

(iii).Rage vs. Fear:

The misclassifications here suggest that certain defensive postures can resemble aggressive behaviors, especially in breeds where raised hackles and a forward-facing stance can indicate both defensive fear and aggressive rage.

### 3.6. Overall Analysis of Our Results

(i)Computational Complexity:

Our contrastive learning approach, specifically using the MoCo framework with modified ResNet18 and ResNet34 encoders, offers a balance between model complexity and performance. By utilizing these lightweight encoders instead of more computationally intensive ones (like ResNet50 or higher), we significantly reduced the overall computational load. Additionally, we experimented with frameworks that adapted to the limited GPU memory, enabling broader accessibility for research with modest hardware.

(ii)Time Efficiency:

The training times across our experimental setups varied depending on the encoder and the specific hyperparameters, such as the batch size and learning rate. For instance, with the ResNet18 encoder on an NVIDIA RTX 3070, the model achieved reasonable convergence within 1200–1800 epochs, with the training times averaging approximately 3.5 h per 1000 epochs. These results are comparable to supervised learning models, but require fewer labeled images, which is a significant advantage in scenarios where large-scale labeled datasets are unavailable or costly to obtain.

(iii)Advantages for Practical Applications:

Our approach achieved approximately 43.2% accuracy, which is promising given the baseline of 14%, while it significantly reduced the need for annotated datasets and achieved faster convergence due to the reduced parameter count in the lighter models. This efficiency positions our framework as a viable solution for real-world applications in canine emotion recognition with modest computational resources.

## 4. Discussion

Some of the significant innovations in data augmentation and random cropping, as well as in color jittering, simulate realistic variability in real-world settings. Increasing the variability further addresses overfitting. Lightweight encoders, such as ResNet18 and ResNet34, were used in efficient training stages without compromising the quality of the learned features, since these models picked the pertinent patterns related to dogs’ facial expressions and body postures. The hyperparameters, such as the batch size and the NT-Xent loss temperature, still affected the performance. Increasing the batch sizes further generally yielded additional negatives to further fine-tune the contrastive representations. The subjective labeling of emotions, along with imbalances in the inherent classes, made training difficult, which was counterbalanced by the inclusion of a KNN classifier to stabilize the embeddings between emotional classes. These factors collectively underscore the deep learning trade-offs we had to navigate to optimize our contrastive learning framework.

Our results show that unsupervised learning can achieve promising results comparable to those of supervised learning. Additionally, using unsupervised learning has the potential to identify new mood categories of dog emotions, beyond the seven emotional feelings of Panksepp [3]. This ability to detect new mood states emerges from the capacity of contrastive learning to find subtle patterns within data without relying on pre-defined labels. In our study, we initially focused on the prediction accuracy of five well-established emotional categories from [5], available publicly on Kaggle [37], as the baseline. By leveraging contrastive learning, a technique that excels in few-shot learning and distinguishing fine-grained differences [39], we demonstrated the model’s ability to recognize and cluster distinct emotional behaviors. Beyond simply categorizing images into predefined moods, contrastive learning could also reveal underlying patterns in dog emotions that may not align with existing categories. For example, clusters of features in the model’s latent space may suggest transitional moods, such as “curious calm” or “playful caution”, which blend attributes from multiple known emotional states. These newly emergent categories could represent moods that dogs experience but are currently unclassified. To identify and validate these potential new mood categories, we could apply clustering algorithms to the model’s latent representations, examining whether specific emotional behaviors form distinct groups that differ from known emotions. Anomaly detection techniques could further isolate behaviors that do not fit existing categories, indicating the need for additional labels. Any proposed categories would then require expert validation, where animal behaviorists and psychologists would assess the distinctiveness and relevance of these newly identified mood states.

The encoders used in this project were standard two-dimensional ResNet CNNs. The encoders were put through pretraining, and were able to produce good accuracies with a trivial downstream classifier, such as a KNN. In the future, for practical applications using this technique, it would be feasible to use a dedicated, fully connected downstream classifier neural network to boost the accuracy scores. However, this was beyond the scope of this preliminary experimental study. With the advent of advanced vision transformer architectures, originally put forth by Dosovitskiy et al. [40], the encoders used for contrastive learning can produce much better results.

This presents a potential pathway to further experimentation by employing transfer learning to fine-tune the model, from its success in emotion recognition in dogs to its application to the identification of emotional expressions in other animals, like cats or horses, or even humans. Testing the model’s cross-species performance would speak to its generalizability and adaptability and possibly reveal shared emotional indicators across different species. Further, such experiments would likely emphasize unique features specific to each species and foster further developments in interdisciplinary research animal behavior, comparative psychology and affective computing. This approach would provide more insights about cross-species emotional communication and, therefore, into applications for animal welfare and human–animal interaction.

### 4.1. Broader Impact on the Field

This research paper on dog emotion recognition using contrastive learning is poised to significantly impact the broader field of animal behavior analysis and artificial intelligence. By introducing a novel dataset comprising images of ten popular dog breeds, each categorized into seven core emotional states, this study provides a robust foundation for future research. This comprehensive dataset, combined with the innovative application of the Momentum Contrast (MoCo) framework for Unsupervised Visual Representation Learning, contributes to the automatic recognition of canine emotions. This paper’s approach not only bridges the gap between supervised and unsupervised learning methodologies, but also demonstrates the feasibility and efficacy of using contrastive learning for emotion recognition in non-human subjects. This has the potential to enhance our understanding of canine emotions, fostering better human–dog interactions and contributing to the well-being of dogs by enabling more precise and empathetic responses to their emotional states.

Furthermore, the broader implications of this research extend to various practical applications, including the development of intelligent systems for dog training, therapy and assistance, where accurate emotion recognition is crucial. By making expert knowledge accessible through AI-driven tools, this study democratizes the ability to understand and interpret dog emotions, benefiting novice pet owners, veterinarians and animal behaviorists alike. The methodological advancements presented in this paper could also inspire similar approaches in the study of other animal species, thus broadening the scope of affective computing and ethology. As this paper integrates cutting-edge AI techniques with ethological insights, it paves the way for interdisciplinary collaborations that could change the way we study and interact with animals, ultimately promoting a deeper and more nuanced understanding of animal cognition and emotions.

### 4.2. Further Scope of This Research

This is also indicative of the prospects of fine-tuning these encoders. They could easily be imported with their current weights, and be fine-tuned to support the identification of the seven Panksepp emotional behaviors [3] in various different domesticated mammalian species. The obvious species which could be studied with minimal fine-tuning and less training data include other four-legged furry animals, such as horses, cats, cows and goats. Another promising future avenue of research could be to explore the transfer learning approach on humans. Dogs and humans have co-existed for several thousand years, which has introduced similarities within our emotional behaviors [41]. This could be demonstrated using the novel perspective of machine learning, by comparing the encoders used for dogs and humans, and analyzing their similarities and differences.

Researchers at the University of California, Berkley, have identified 27 categories of emotions in humans [42]. Currently, canine emotions have not been explored in extensive depth, and our research aims to lay down a foundation on which extensive emotional research could be carried out on dogs, with a new perspective and assistance from the cutting-edge methodologies developed in machine learning. The global population of dogs is steadily increasing, with estimates ranging from 900 million to one billion dogs in 2024 [43]. This calls for an increase in the overall academic research conducted on canines. Our research aims to add to animal and computer interactions, to aid this research carried out on dogs.

As previously outlined, our study utilizes a unique dataset consisting of images of ten popular dog breeds. While this provides a solid foundation for the model’s evaluation, it limits the model’s ability to generalize to less common or mixed-breed dogs. This narrow breed selection may reduce the model’s robustness when applied to a broader range of canine phenotypes. Future research could focus on expanding the dataset to include a more diverse range of breeds, including mixed-breed dogs, to better assess the model’s generalizability and robustness in real-world scenarios.

Considering this, we chose the MoCo framework for its superior unsupervised learning method exceling at capturing subtle differences in image representations, especially in tasks that require fine-grained visual distinction, such as emotion recognition. Autoencoders are also effective for learning compressed representations, but tend to focus more on reconstruction and less on the feature discrimination that is needed to separate complex emotional expressions. Generative adversarial networks are powerful when generating new data but are, in general, less effective at extracting meaningful, clustered embeddings for downstream classification tasks because the optimization performed is for realism rather than the separation of features in an embedding space. Instead, this MoCo framework uses contrastive learning to map similar instances closer together and push dissimilar instances apart in the latent space, hence creating more unique clusters for each emotional state. Preliminary experiments with Autoencoders and GAN-based feature extraction had lower accuracies and less discriminative emotional embeddings compared to MoCo, which justified our choice further. On the whole, MoCo gave us the best balance of computational efficiency and the separability of features for our unsupervised emotion recognition dataset.

### 4.3. Ethical Considerations

In our pursuit of advancing the understanding of canine emotions through contrastive learning, we must also consider the ethical implications of this research. Consequently, we adhere to the welfare-centered ethics framework introduced by [44] and the guidelines from [45] that address ethical research in the ACI discipline.

Our research primarily focuses on improving animals’ life quality [44] by enhancing the understanding of dogs’ feelings, thus enabling the more accurate handling of them. Additionally, we aim to foster interspecies relationships [44] by improving the mutual understanding and communication between humans and animals. Throughout this process, we ensure animals’ welfare [46] at all times. Here, we outline the ethical considerations made before engaging in this research.

One of the primary ethical considerations revolves around the welfare of the animals involved. To ensure this, we designed our data collection process [46] to minimize any potential distress or harm to the dogs [46]. We conducted photographing and filming sessions in the dogs’ natural or familiar environments, preferably performed by their owner, avoiding any intrusive or stressful interventions [46].

Another critical ethical concern is the potential misuse of the technology developed from this research. To address this, we established clear guidelines and regulations to govern the application of this technology. These measures were designed to prevent the exploitation of the technology in ways that could harm animals, such as the inappropriate manipulation or commercialization of their emotions for entertainment purposes [46]. By implementing these guidelines, we ensure that the technology is used responsibly and ethically, enhancing the welfare and understanding of dogs rather than serving exploitative interests.

Lastly, we address the ethical considerations related to the privacy and data protection of the pet owners and their dogs. We implemented strict measures to handle potentially sensitive data, such as images and videos of the dogs [47]. Before collecting any data, we obtained informed consent from the pet owners, clearly explaining the purpose, methods and potential uses of the research. We anonymized the data to prevent the identification of individual dogs or owners and implemented secure storage protocols to safeguard the information from unauthorized access. As a result, we are unable to make our dataset publicly available.

By addressing these ethical concerns, our research contributes positively to the field of canine emotion recognition and the research field of ACI while maintaining the highest standards of ethical integrity.

## 5. Conclusions

We presented a contrastive learning approach for detecting the seven Panksepp emotions in dogs based on their body posture and facial emotions using Momentum Contrast for Unsupervised Visual Representation Learning. We achieved an accuracy of 43.2%, surpassing the 14% baseline on our self-curated dataset. We were able to draw conclusions about our model by comparing it with supervised approaches applied to our dataset and the publicly available Kaggle dataset. Our approach offers valuable insights into dog emotion recognition without having to rely on labeled data. It also uses less parameters compared to supervised learning. Overall, our study demonstrates the feasibility of using unsupervised learning techniques for dog emotion recognition, providing a promising avenue for increasing our understanding of the emotional world of humans’ best friend. View the Appendix A for reference.

## Figures and Tables

**Figure 1 sensors-24-07324-f001:**
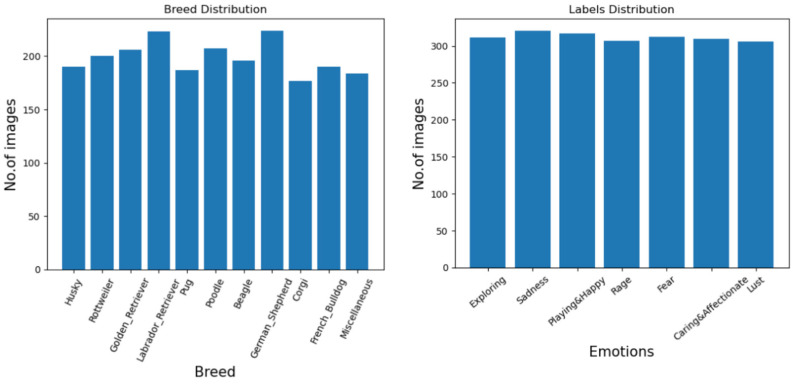
Breed and emotion distribution of images in dataset.

**Figure 2 sensors-24-07324-f002:**
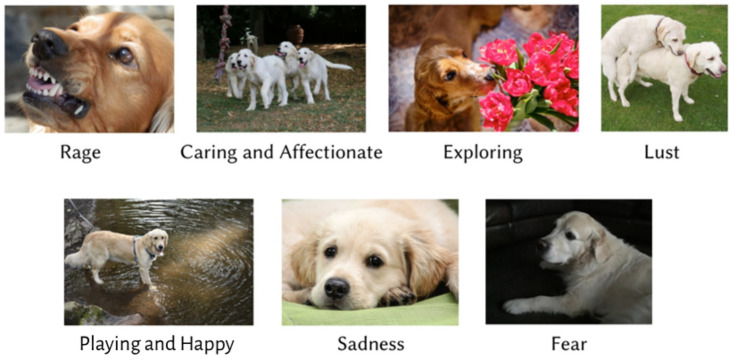
Examples of seven emotional behaviors for Golden Retriever breed.

**Figure 3 sensors-24-07324-f003:**
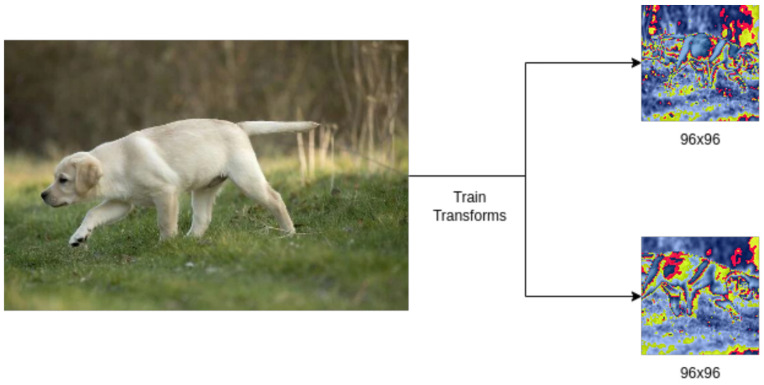
An image of a labrador puppy displaying the ‘Exploring’ behavior augmented according to the above-mentioned augmentations (image source: Wikipedia).

**Figure 4 sensors-24-07324-f004:**
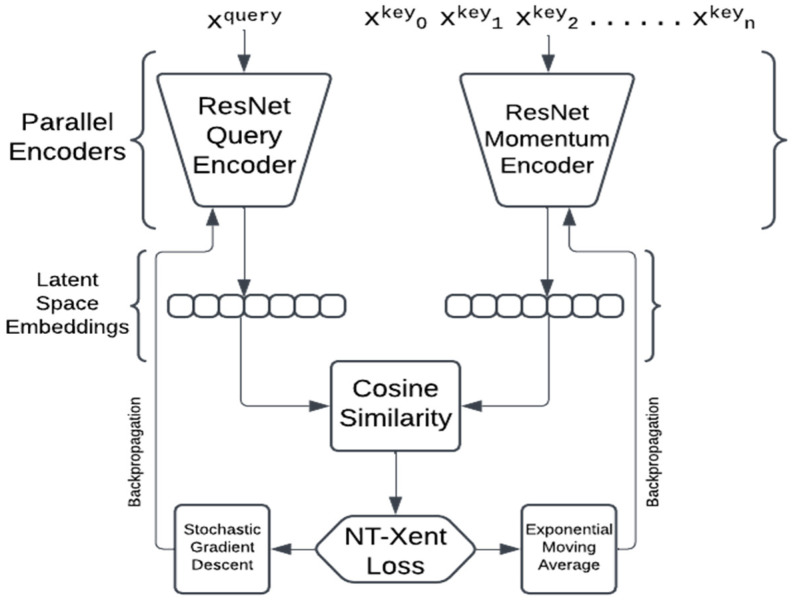
Contrastive learning MoCo framework for dog emotion recognition using ResNet encoders.

**Figure 5 sensors-24-07324-f005:**
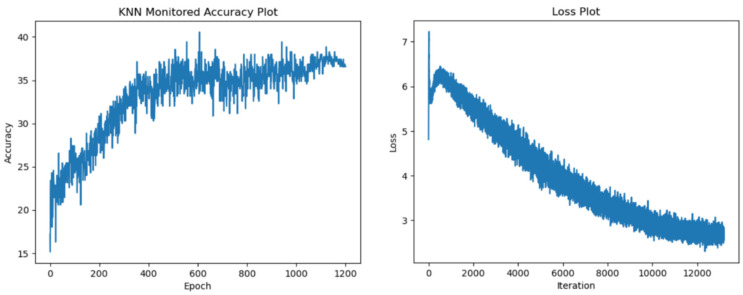
Test accuracy and training loss results for ResNet18 encoder with 1200 epochs.

**Figure 6 sensors-24-07324-f006:**
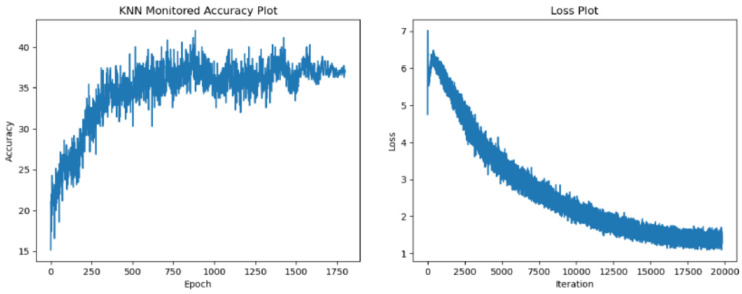
Test accuracy and training loss results for ResNet18 encoder with 1800 epochs.

**Figure 7 sensors-24-07324-f007:**
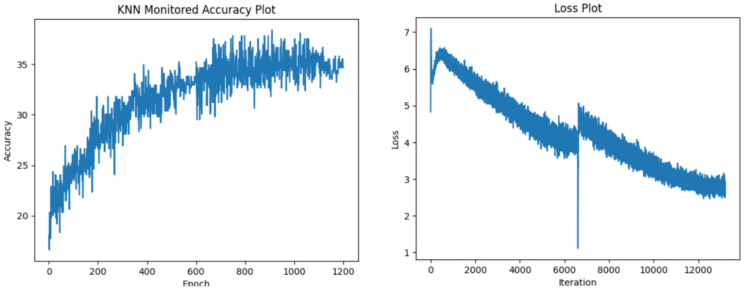
Test accuracy and training loss results for ResNet34 encoder with 1200 epochs.

**Figure 8 sensors-24-07324-f008:**
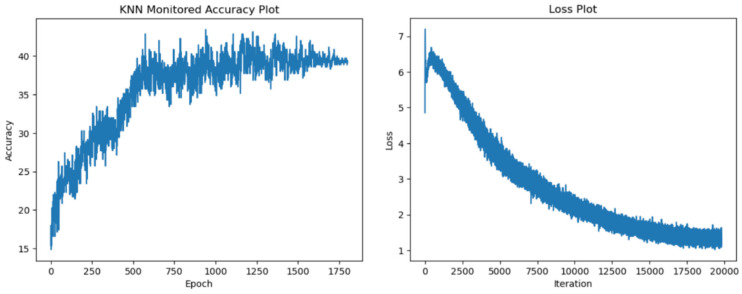
Test accuracy and training loss results for ResNet34 encoder with 1800 epochs.

**Figure 9 sensors-24-07324-f009:**
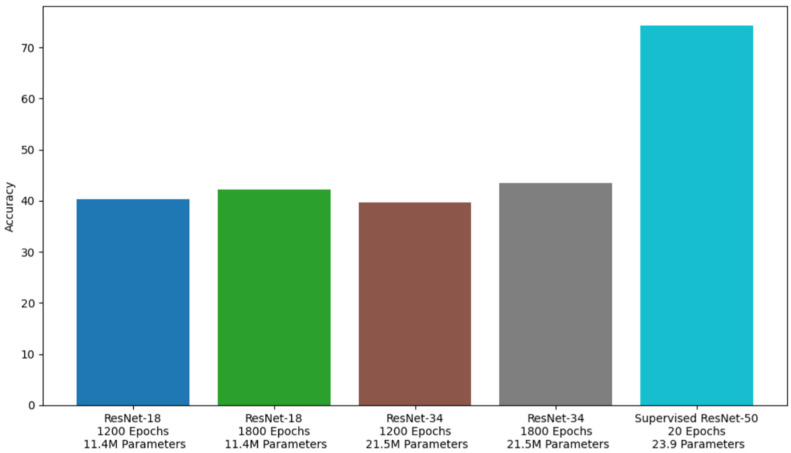
Unsupervised and supervised model comparison.

**Table 1 sensors-24-07324-t001:** Best four results produced by MoCo frameworks.

Run	Image Resolution	Encoder	GPU	Batch Size	LR	K Value KNN	Momentum	Epochs	NT-Xent Temperature	Accuracy
1	96	ResNet18	NVIDIA RTX-3070	256	0.3	150	0.99	1200	0.1	40.24%
2	96	ResNet18	NVIDIA RTX-3070	128	0.3	200	0.99	1800	0.1	42.19%
3	96	ResNet34	NVIDIA Tesla P100	128	0.25	150	0.95	1200	0.1	39.71%
4	96	ResNet34	NVIDIA Tesla P100	128	0.3	200	0.99	1800	0.1	43.42%

**Table 2 sensors-24-07324-t002:** Comparison of accuracy of ResNet50 to accuracy of MoCo results on our own dataset.

Emotion	Accuracy ResNet50	Accuracy MoCo
Caring	94.74%	34.61%
Exploring	83.75%	40.40%
Fear	28.95%	35.51%
Lust	47.05%	62.79%
Playing	46.34%	38.88%
Rage	87.09%	45.91%
Sadness	78.57%	44.37%

**Table 3 sensors-24-07324-t003:** Further classification results of ResNet50 model on our own dataset.

Emotion	Precision	Recall	F1-Score
Caring	0.9796	0.7059	0.8205
Exploring	0.7609	0.6364	0.6931
Fear	0.5156	0.9429	0.6667
Lust	0.8125	0.7091	0.7573
Playing	0.7778	0.9403	0.8514
Rage	1.0000	0.7037	0.8216
Sadness	0.7600	0.5352	0.6281

**Table 4 sensors-24-07324-t004:** Comparison of accuracy of ResNet50 model to accuracy of MoCo results on publicly available dataset [37].

Emotion	Accuracy ResNet50	Accuracy MoCo
Angry	81.08%	55.50%
Happy	96.47%	45.65%
Relaxed	91.13%	35.90%
Sad	80.00%	54.55%

**Table 5 sensors-24-07324-t005:** Further classification results of ResNet50 model on publicly available dataset [37].

Emotion	Precision	Recall	F1-Score
Caring	0.9796	0.7059	0.8205
Exploring	0.7609	0.6364	0.6931
Fear	0.5156	0.9429	0.6667
Lust	0.8125	0.7091	0.7573

## Data Availability

Data are contained within the article.

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
