# Peer review of "Unsupervised Canine Emotion Recognition Using Momentum Contrast"

_sensors, 2024, doi:10.3390/s24227324_

Round 1
Reviewer 1 Report
Comments and Suggestions for Authors
This paper presents a study on unsupervised canine emotion recognition using the Momentum Contrast framework, focusing on identifying emotions based on dogs' facial expressions and body posture. The authors constructed a dataset of 2184 images across ten dog breeds, categorized into seven emotional states by Jaak Panksepp. They modified the MoCo framework for unsupervised learning, achieving an accuracy of 43.2% on their dataset, which was compared to a supervised ResNet50 model that achieved 74.32% accuracy. The study suggests that unsupervised learning can identify dog emotions effectively without labeled data. It also discusses the broader implications for animal behavior analysis and AI, as well as ethical considerations regarding animal welfare and data privacy. The conclusion highlights the feasibility of using unsupervised learning techniques for understanding the emotional world of dogs. Overall, the objective of this work is clear and interesting, and the proposed technique sounds available. However, several issues should be improved, as listed below:
1. This manuscript uses many acronyms and terms. I think it is necessary to add a table regarding their full names and usage. It will be more convenient.
2. The dataset was sourced from Google Images and Wikipedia. What measures were taken to ensure the images were representative and not biased towards certain breeds or emotions?
3. I suggest the authors should include the figures regarding the overall framework and network architecture in suitable places in the manuscript.
4. The supervised learning model is based on ResNet50. Could the authors show more details about the choice of this architecture over other potential models in this field and how it was adapted for such a specific task?
5. What is the tool used for programming in this work? Matlab? Python? Please include the conditions of the experiments, e.g., OS, CPU, GPU, etc.
6. The authors should provide details concerning the overall complexity and time consumption in the results, which can further demonstrate the advantages of this study.
7. In Section 4, could the authors discuss which factors had the most significant impact on model performance and why they believe this is the case through a deep learning view?
8. The authors mention the potential for the models to identify new mood categories. What these new categories might be, and how they might be identified based on the proposed method?
9. Finally, the authors also mention the potential for using the encoders with other mammalian species. It will be better to elaborate on how they envision this transferability and what additional challenges might be encountered.
Comments on the Quality of English LanguageThere is not much issue regarding the language aspect, but the academic writing standards for a good paper need to be paid attention to. For example, when abbreviations first appear, their full names should be marked. Please refer to the high-quality papers in this field and then improve this paper.
Reviewer 2 Report
Comments and Suggestions for Authors
This manuscript sensors-3277977 presents an innovative approach for unsupervised canine emotion recognition utilizing the Momentum Contrast (MoCo) framework. The authors aim to develop a system capable of identifying dog emotions based on facial expressions and body posture. To achieve this, they constructed a comprehensive dataset comprising 2184 images of ten popular dog breeds, categorizing these images into seven emotional labels proposed by Jaak Panksepp. These labels include 'Exploring', 'Sadness', 'Playing', 'Rage', 'Fear', 'Affectionate', and 'Lust'. The authors modified the MoCo framework to train on this dataset, achieving an accuracy of 43.2%, which significantly outperforms the baseline accuracy of 14%. Additionally, they tested their model on a publicly available dataset, attaining an accuracy of 48.46% against a baseline of 25%. The authors further compared their unsupervised approach with a supervised model using the ResNet50 architecture, demonstrating that the supervised model achieved a higher accuracy of 74.32% on their dataset. Nonetheless, the paper demonstrates the feasibility and potential of unsupervised learning techniques for canine emotion recognition. It was a pleasure reviewing this work and I can recommend it for publication in Sensors after a major revision. I respectfully refer the authors to my comments below.
1. The English needs to be revised throughout. The authors should pay attention to the spelling and grammar throughout this work. I would only respectfully recommend that the authors perform this revision or seek the help of someone who can aid the authors.
2. (Section 1 Introduction) The reviewer hopes the introduction section in this paper can introduce more studies in recent years. The reviewer suggests authors don't list a lot of related tasks directly. It is better to select some representative and related literature or models to introduce with certain logic. For example, the latter model is an improvement on one aspect of the former model. Please delete the DOI number in the references.
3. The paper mentions the construction of a novel dataset comprising images of ten popular dog breeds. However, it would be interesting to see how the model generalizes to less common breeds or mixed-breed dogs. Including a more diverse set of dogs in the dataset could provide insights into the model's robustness.
4. The authors have focused primarily on the MoCo framework. Comparing the performance of MoCo with other unsupervised learning methods, such as Autoencoders or Generative Adversarial Networks (GANs), would strengthen the evaluation and demonstrate the superiority of the chosen approach.
5. Providing a detailed analysis of the misclassifications made by the unsupervised model could offer valuable insights into the challenges of unsupervised canine emotion recognition. This could guide future improvements in model architecture or data preprocessing steps.
6. The current approach relies solely on facial expressions and body posture. Incorporating additional contextual information, such as the dog's environment or the presence of other animals or humans, may improve the model's accuracy. Experiments exploring the impact of such contextual cues would be insightful.
7. Since the authors mention the potential for transfer learning, conducting experiments where the model trained on dog emotions is fine-tuned for other animal species or even human emotions would demonstrate the model's generalizability. This could open new avenues for interdisciplinary research.
8. While the authors briefly discuss ethical considerations, a more detailed discussion on the measures taken to ensure the welfare of the dogs during data collection would be beneficial. This includes aspects such as informed consent from dog owners and minimizing stress during photo sessions.
9. The paper mentions using various data augmentation techniques. A more thorough ablation study analyzing the impact of each augmentation technique on model performance would provide a better understanding of their effectiveness.
10. The paper concludes by mentioning potential practical applications of the research. Expanding on these ideas and discussing how the model could be integrated into real-world systems, such as smart dog collars or training aids, would make the work more impactful and applicable.
My overall impression of this manuscript is that it is in general well-organized. The work seems interesting and the technical contributions are solid. I would like to check the revised manuscript again.
Comments on the Quality of English Language
The English needs to be revised throughout. The authors should pay attention to the spelling and grammar throughout this work. I would only respectfully recommend that the authors perform this revision or seek the help of someone who can aid the authors.
Round 2
Reviewer 1 Report
Comments and Suggestions for Authors
The authors have addressed my queries, and their responses have been convincing. As a result of their revisions, the quality of the manuscript has improved. Therefore, I recommend it for acceptance.
Reviewer 2 Report
Comments and Suggestions for Authors
The revised manuscript is improved compared to the former version. My previous comments are well addressed, and the presentation is improved significantly. The composition pattern and some other ideas are well elaborated, making them clearer. Overall, I tend to accept this manuscript.